# eHealth Platforms Facilitate Prostate Cancer Shared Care: A Systematic Review

**DOI:** 10.3390/healthcare12171768

**Published:** 2024-09-04

**Authors:** David C. Homewood, Jodie Mcdonald, Zina Valaydon, Cindy Ogluszko, Olga A. Sukocheva, Edmund Tse, Niall M. Corcoran, Guru Iyngkaran

**Affiliations:** 1Department of Urology, Western Health, Melbourne, VIC 3020, Australia; niallmcorcoran@gmail.com; 2Royal Melbourne Hospital, Melbourne, VIC 3052, Australia; giyngkaran@yahoo.com.au; 3Department of Gastroenterology, Western Health, Melbourne, VIC 3020, Australia; jodie.mcdonald@wh.org.au (J.M.);; 4Department of Urology, St Vincent’s Hospital, Melbourne, VIC 3065, Australia; prostatecancernurse@wh.org.au; 5Department of Gastroenterology and Hepatology, Royal Adelaide Hospital, Adelaide, SA 5000, Australia; edmund.tse@sa.gov.au; 6Department of Surgery, University of Melbourne, Parkville, VIC 3052, Australia; 7Victorian Comprehensive Cancer Centre, Melbourne, VIC 3052, Australia

**Keywords:** digital health, digitalised care, eHealth platform, prostate cancer survivorship, symptom tracking

## Abstract

Background: Prostate cancer survivorship care is essential for the early identification of cancer recurrence and progression and the monitoring of adverse effects. Prostate cancer survivorship programs have enabled care to be shared between specialists using digital healthcare platforms. We systematically reviewed the literature to examine if prostate cancer survivorship care had been successfully digitalised. Methods: English language articles were searched on PubMed, Embase, and Cochrane Libraries. The search terms included combinations of “eHealth”, “digital health”, “prostate cancer”, “shared care”, and related keywords (studies published between [1 January 1946 and 20 March 2023]). Results: Our search strategy yielded 1722 publications, of which 17 studies were included in our final review. Diverse eHealth interventions (web platforms, apps, patient portals) for digital prostate cancer shared care enabled communication, symptom management, and holistic assessment, with potential for reducing anxiety, enhancing outcomes, and increasing engagement. The studies (9 months to 5 years duration) involved participants across different care phases (16 to 3521 participants). We identified ten eHealth platforms, which provided successful symptom tracking, needs assessment, and communications. The platform-based interventions improved some aspects of communication, symptom management, and care delivery. The ongoing clinical need for a robust digital platform that caters to all domains of shared care was identified. Conclusions: eHealth will certainly play a central role in digital prostate cancer shared care, providing better health outcomes and care delivery. Future larger studies in this field should address the implementation barriers, including cost-effectiveness and primary care remuneration. It is also crucial to refine application useability and workflow, focusing on standardization and patient-centred approaches.

## 1. Introduction

Prostate cancer is the prevalent internal malignancy in men, with 1.4 million men globally and 460,000 men in Europe diagnosed in 2020 [1,2,3]. Survival rates post-diagnosis have doubled (from 1993–2013), with 5-year survival ranging from 97.1% [1] in the USA to 83% in Europe [2,3]. More than 3.3 million prostate cancer survivors exist in the USA alone. This is set to increase with earlier detection and more effective treatments, straining survivorship care demand [1].

Vigilant monitoring is crucial to detect disease recurrence, treatment side effects, and psychological distress. Key areas of prostate survivorship care include management of incontinence, erectile dysfunction, and mental health problems [4]. Androgen deprivation therapy further increases risks of metabolic disease, heart issues, and mood disturbance [5]. With increasing diagnosis and survival, the survivorship population continues to expand, further stressing outpatient services and emergency departments. This issue is amplified in remote areas by geographical barriers, thus necessitating innovation in delivery of shared care [6].

Shared survivorship care has been defined as the joint participation of primary and specialist clinicians in planned delivery of care informed by an enhanced information exchange [7,8]. Prostate cancer shared care models involving urologists, nurses, and family doctors have demonstrated success [9]. However, effective coordination requires significant time and financial investment [4,10]. One American study used SEER-Medicare data (79,826 men categorized by costs, from 1992–2005, ages 66–99, ~14% of survivorship population) to quantify these factors [10]. Whilst initial care incurred the highest cost (AUD 987 million), continuing care (AUD 533 million) was a significant expense (per patient yearly cost AUD 1464). Notably, after 8 years, continuing care costs surpassed initial care for survivors (AUD 10,248 vs. AUD 8799). Of this, office visits for follow-up care comprised 27.3% (the most significant cost was androgen deprivation therapy at 62.7%). This ongoing survivorship cost is a significant public health issue. eHealth technologies have been proffered as a solution to this; efficiently coordinating multidisciplinary, multi-institutional care reducing the need for outpatient review [6,11]. We systematically reviewed the literature to explore eHealth’s emerging role in digital prostate cancer shared care, interrogating the limitations of available technology and the implementation barriers they face.

## 2. Methods

### 2.1. Research Question and Search Strategy

The primary research question guiding this systematic review is as follows: How does eHealth contribute to prostate cancer shared care?

A systematic and comprehensive search was conducted across electronic databases, including PubMed, Embase, and Cochrane Library. Study eligibility was assessed using the ‘Patient, Intervention, Comparison, Outcome’ (PICO) basis. We included the studies that targeted patients with prostate cancer and/or prostate cancer survivors. The main interventions included the application of a digital healthcare platform and the assessment of the outcome of their implementation. Comparison to the paper-based system of shared care was considered but not always implemented. Various forms of desirable outcome were assessed, including cost-effectiveness/minimization, improvement of care, and/or health.

The search terms included combinations of “eHealth”, “digital health”, “prostate cancer”, “shared care”, and related keywords. The search was limited to studies published between [1 January 1946 and 20 March 2023] (Figure 1). Additionally, references from relevant articles, grey literature including abstracts, conference procedures, and online tools were hand-searched to identify any additional studies. All selected studies were screened by two separate authors (DH and JM), and disagreements were independently adjudicated (GI).

This systematic review protocol and description of the search strategy were registered and published in the PROSPERO database on 18 July 2024. The project registration ID is CRD42024566664.

To identify all steps in the selection process and analysis, the authors adapted the Preferred Reporting Items for Systematic Reviews and Meta-Analyses (PRISMA) checklist (Appendix A) which is designed to explain the process of gathering and selection of the appropriate information.

### 2.2. Inclusion Criteria

Full-text English original articles focusing on eHealth ‘shared care’ interventions in prostate cancer shared care were eligible. Full text studies which described care model or trial-based evaluations were included. Papers that were not presented in full texts (e.g., abstracts) and non-research studies (e.g., reviews) were excluded, although some of them were screened to identify potential links to the full text originals (Figure 1). Duplicate manuscripts were excluded using Covidence application tool (https://www.covidence.org/; accessed on 5 September 2023).

### 2.3. Shared Care Application Definition

This review focused on shared care applications, which were defined as including evidence-based Patient-Reported Outcome Measures (PROMs) (at more than two discrete timepoints including: patient satisfaction, quality of life, and clinical outcomes), communication tools (either instant message, email notification, or asynchronous text based), as well as clinician-led intervention rather than solely application-led self-care symptom control. These studies were chosen as they best addressed the key domains of shared care.

### 2.4. Risk of Bias Assessment (Quality Evaluation)

The selected articles were evaluated using the Strengthening the Reporting of Observational Studies in Epidemiology (STROBE) tool to estimate the quality of the published data. The sources of information, design of the study, and the description of the sample (number of participants and centres) were assessed for each full-text article. 

## 3. Results

The studies examined included various eHealth interventions, such as web-based platforms, smartphone applications, and multi-platform patient portals. Key findings indicated that these interventions facilitated varying amounts of communication, symptom management, and holistic needs assessment.

### 3.1. Study Selection and Characteristics

Seventeen studies were selected for inclusion in the systematic review, focusing on eHealth interventions in digital prostate cancer shared care (Table 1). These studies were chosen based on keyword-based search strategy and predefined inclusion criteria.

The selected studies employed a variety of study designs, including non-randomized controlled feasibility studies [12], randomized controlled trials [13,14,15,16,17], prospective observational studies [12,18,19], and multi-centre cluster-randomized stepped wedge trials [20]. The duration of the studies ranged from 9 months [16] to 5 years [12], with some studies still ongoing [20].

### 3.2. Participant Characteristics

The studies included diverse populations of prostate cancer patients, encompassing various phases of survivorship cancer care, including radiotherapy, chemotherapy, and post-surgical care. Sample sizes varied across studies, ranging from 16 to 3521 participants, reflecting the heterogeneity of the patient populations under investigation (Table 1).

### 3.3. eHealth CaP Shared Care Platform Characteristics

There were 10 discrete eHealth platforms applicable to prostate cancer share care identified. These were as follows: Composite Holistic Needs Assessment Adaptive Tool-Prostate (CHAT-P) (1 protocol, 1 study) [12,21], e-OncoNote (1 study) [13], PROMIS CAT T (1 study) [19], eSYM (1 protocol) [20], Interaktor (6 studies) [14,15,22,23,24,25], sHNA (1 study) [16], PERC (Patient Education Resources for Couples) (1 study) [26], RyPros (1 study) [18], TrueNTH (1 study) [27], and WebChoice (1 protocol, 1 study) [17,28]. The eHealth interventions examined in the studies showcased a wide range of functionalities and features tailored to support shared care in prostate cancer (Table 2). These interventions included holistic needs assessment tools [16], symptom-tracking applications, patient-reported outcome measure surveys, communication platforms between primary care providers and specialists [13,15,18,19,20], and web-based programs for patient education and support [16]. Adaptive tools and real-time alerts were also incorporated in some interventions to enable personalized care delivery [16,17,18]. The extent to which these tools fulfilled the demands of shared care were colour-coded with a traffic light system (green: effective; orange: somewhat effective; red: not effective) (Table 2).

### 3.4. Intervention Characteristics

A variety of eHealth interventions have been used to improve communication, symptom management, and overall care delivery across these platforms. The heterogeneity of the selected studies was noted and discussed as part of the qualitative assessment.

#### 3.4.1. Holistic Needs Assessment (HNA) Tools

CHAT-P, a web-based shared care platform, was built around the principals of HNA [12,21]. This includes patient-reported outcome measures, information provision, and support for the financial and legal aspects of care. The rationale of HNA tools was to create a more holistic understanding of patients’ physical, psychological, and social needs, enabling personalized care planning. The published pilot study was limited to qualitative interview data about perceived benefits around time saving and holistic care. However, at this stage, there has been no proven quantified benefit from this platform.

#### 3.4.2. Symptom Tracking and Management

Various interventions focused on symptom tracking and management throughout the prostate cancer care journey. Smartphone applications, such as Interaktor, enabled patients to perform daily symptom scoring and receive real-time alerts based on the occurrence, frequency, and distress of symptoms [14,15,22,23,24,25]. Interaktor then provided evidence-based self-care advice and facilitated targeted communication between patients and healthcare providers. By empowering patients to monitor and manage their symptoms, these interventions aimed to improve symptom control and overall well-being.
healthcare-12-01768-t001_Table 1Table 1Summary of shared care applications in prostate cancer and their key characteristics.TitleAuthor (Year), Journal Study Design Study Population eHealth Shared Care Modality eHealth InterventionComparator OutcomesDuration ConclusionIntegrated Care in Prostate Cancer (ICARE-P): Non-randomized Controlled Feasibility Study of Online Holistic Needs Assessment, Linking the Patient and the Health Care Team [21]Nanton et al. (JMIR), 2017Protocol—Non-randomized Controlled Feasibility StudyN/AWeb-based, adaptive, cancer-specific needs assessmentComposite Holistic Needs Assessment Adaptive Tool-Prostate (CHAT-P)—seeks to provide a holistic needs assessment (HNA) ranging from patient-reported outcome measures to information provision to financial and legal aids. It was used as a method for PROM, education, and communication adjunctNilN/A9 months N/AA Web-Based Prostate Cancer‚ Specific Holistic Needs Assessment (CHAT-P): Multimethod Study From Concept to Clinical Practice [12]Nanton (2022), JMIRMultimethod Study16 patients from 2 outpatient clinics. 5 from site 1, 11 from site 2. RP 3, AS 8, RT 2, Unknown 3. No comparator.NilNo clinical outcomes. Qualitative data on application usability only: (user interface and design, suitability of content, personal value, and implementation and use in the clinical care pathway5 years First web-based interactive platform for cancer-specific HNA. No clinical reported outcomes.Web-Based Asynchronous Tool to Facilitate Communication Between Primary Care Providers and Cancer Specialists: Pragmatic Randomized Controlled Trial [13]Petrovic (JMIR), 2023Pragmatic randomised controlled trial173 patients randomised to either interventiongroup (eOncoNote + usual inter-clinician communication) or control. 104 (60.1%) patients in the survivorship phase (breast and colorectal cancer) and 69 (39.9%) patients in the treatment phase (breast and prostate cancer).Web- and text-based asynchronous systemeOncoNote—designed to facilitate communication between care providersControl group (usual communication only), including 104 (60.1%) patients in the survivorship phase (breast and colorectal cancer) and 69 (39.9%) patients in the treatment phase (breast and prostate cancer)Primary outcome: patient-reported team and cross-boundary continuity (Nijmegen Continuity Questionnaire) (no significant difference)Secondary outcome: Generalized Anxiety Disorder Screener (GAD-7), Patient Health Questionnaire on Major Depression, and Picker Patient Experience Questionnaire12 months eOncoNote had no significant effect on patient-reported continuity of care, but reduced patient anxiety at long-term follow-up (GAD-7 *p* < 0.004)Implementing Electronic Health Record‚ Integrated Screening of Patient-Reported Symptoms and Supportive Care Needs in a Comprehensive Cancer Centre [19]Garcia et al. (2019), CancerProspective observational study 3521 oncology patients (51.6%) completed 8162 assessments (PROMIS CAT T); approximately 55% of the responding patients completed 2 or more surveysPatient portal PROMIS CAT T (EPIC, MyChart)This oncology study integrated a custom-built patient-reported outcome measure survey (PROMIS CAT T) into an existing EHR (Epic). 51.6% of patients enrolled completed 8162 assessments, identifying prevalent symptoms and care needsComputer-scored and benchmarked on the basis of normative data from patients with cancer and the general populationPatient endorsement of supportive care needs was associated with significantly higher anxiety, depression, fatigue, and pain interference scores and lower physical function scores. Patients who triggered clinical alerts tended to be younger and more recently diagnosed, to have greater comorbidities, and to be a racial/ethnic minority. Patients who triggered clinical alerts had more healthcare service encounters in the ensuing month32 months (January 2015–August 2017)EHR integration enabled efficient PROM and facilitated clinical review when clinical alerts activated.Implementation of patient-reported outcomes for symptom management in oncology practice through the SIMPRO research consortium: a protocol for a pragmatic type II hybrid effectiveness-implementation multi-center cluster-randomized stepped wedge trial [20]Hassett et al. (2022), TrialsMulti-center cluster-randomized stepped wedge trialOncology and surgical oncology patients (planned)Smartphone ApplicationEpic patient portal with novel eSyMeSyM supports real-time symptom tracking for patients, automated clinician alerts for severe symptoms, and specialised reports to facilitate population management.Patients who started chemotherapy orhad surgery before eSyM deployment (i.e., control epi-sodes)Primary Outcome: ED treat-and-release event within 30 days of starting chemotherapy or being discharged following surgery.Secondary outcomes: include hospitalization rates, chemotherapy use (time to initiation and duration of therapy), and patient quality of life and satisfactionOn-going (approved in 2018; due to finish end of 2023)N/AEarly detection and management of symptoms usingan interactive smartphone application (Interaktor) during radiotherapy for prostate cancer [14]Sundberg (2016), Support Care CancerTwo centre non-randomized controlled study in Sweden 130 patients undergoing radiotherapy for localized prostate cancer, 64 (control group) and 66 (intervention group)Interactive Mobile Application for smartphone/ tabletInteraktor offers daily symptom scoring with a 14:00 reminder for incomplete forms. It provides self-care advice and alerts nursing staff for targeted patient contact during business hours.The system includes patient symptom assessment, a web interface for monitoring, a risk assessment model sending alerts to healthcare providers, evidence-based self-care guidance, and symptom history graphs.It tracks 14 symptoms on weekdays, post-radiotherapy. Researchers provided usage instructions, self-care advice, checklists, and tech support.Self-reports went to study nurses via a secure interface64 (control group)—historically controlled: balanced regarding demographics and clinical characteristics, except that the CG showed a statistically significant lower level of education. At baseline (T1), there were no statistically significant differences between the IG and the CG regarding any of the outcome measures.- 3 surveys EORTC QLQ-C30 at: T1 (treatment initiation), T2 (treatment completion), and T3 (3 months post completion).Radiotherapy of varying duration (5–8 weeks—EBRT +/− HDR~6 months)Interaktor group had significantly lower levels of fatigue, nausea, insomnia, and urinary symptoms at end of treatment and 3 months post. mHealth tools aid facilitating supportive care needs during cancer treatment and contribute to improved patient outcomes.Adherence to Report and Patient Perception of an Interactive App for Managing Symptoms During Radiotherapy for Prostate Cancer: Descriptive Study of Logged and Interview Data [24]Langius-Ekl√∂f et al. (2017), JMIR CancerDescriptive Study 66 patients with prostate cancer receiving radiotherapyNilPrimary Outcome: Not reported. Inferred to be adherence to reporting symptoms daily: 87% (median 92%, range 16%–100%)Between 56 and 77 days Use of a shared care application (Interaktor) increased patients’ sense of security, feeling of well-being and ability to self-manage symptoms.Patients’ Perspective on Participation in Care With or Without the Support of a Smartphone App During Radiotherapy for Prostate Cancer: Qualitative Study [23]Nyman (2017), JMIR MHEALTH AND UHEALTHMulticentre RCT, Qualitative 28 prostate cancer patients, 17 intervention group (smartphone app), 11 control (standard care)11 patients in control group (standard care)Primary Outcome: Not stated. Main themes explored in results: mutual participation (improved in intervention), fight for participation, requirement for participation, and participation in getting basic needs satisfiedNot specifiedInteractive apps for symptom reporting enable targeted real-time health provider contact and can increase patient participation in care.Effects of an interactive mHealthinnovation for early detection of patient-reported symptom distress with focus onparticipatory care: protocol for a studybased on prospective, randomised, controlled trials in patients with prostateand breast cancer [25]Langius-Ekl√∂f et al. (2017) BMC CancerProtocol RCT for Interaktor 150 prostate cancer patients Standard treatment and care consist of neo adjuvant chemotherapy and regular visits to the physician and the oncology contact nurse prior to every treatment occasionOutcomes concerning HRQoL, symptom distress, perception of individual care, sense of coherence and health literacy (EORTC-QLC-C30, Memorial Symptom Assessment Scale (MSAS), Individual Care scale (ICS))18 weeks for breast cancer, 9 weeks for prostate cancerN/AEngagement in an Interactive App for Symptom Self-Management during Treatment in Patients With Breast or Prostate Cancer: Mixed Methods Study [15]Crafoord (2020), Journal of Medical Internet ResearchTwo separate randomized controlled trials Two distinct populations: 149 (74 intervention) patients receiving neoadjuvant chemotherapy for breast cancer and 146 (73 intervention) patients receiving radiotherapy for prostate cancerControl group (standard care 75 breast, 73 prostate)Primary Outcome: Not reported. Inferred to be adherence: 83%High patient engagement with application (83% daily symptom scoring), feeling of added support and safety, easy method of direct contact with healthcare professionals.Patients’ Individualized Care Perceptions and HealthLiteracy Using an Interactive App During Breast andProstate Cancer Treatment [22]Crafoord et al. (2023), Computer Informatics NursingControl group (standard care 75 breast, 73 prostate)Primary outcome: Individualized Care ScaleControl group rated individuality in the care delivered lower regarding decision control (ICS-B, Dec-B) comparedintervention group at follow-up (*p* = 0.041 effect size of 0.4. No other differences between the interventionand control groups were observed.For patients treated for prostate cancer, application use had a moderately positive effect on their perception of individualisation in care, decision control, and ability to find and understand health information. However, no such effects were observed in patients treated for breast cancer.Promoting integrated care in prostate cancer through online prostate cancer-specific holistic needs assessment: a feasibility study in primary care [16]Clarke (2019), 9-month non-randomised cluster controlledfeasibility study14 general practices (8 intervention and 6 control), and 41 men (29 intervention and 12 control)Digital online PlatformOnline prostate cancer-specific holistic needs assessment (sHNA) 3 times over 9 months and shared digital communication12 control patients—standard care Primary Outcome:Feasibility and acceptability—Technology Acceptance Model (TAM) questionnaire returned by 11/29 patients (mainly acceptable—common issues logging in)Patient-reported outcome measures—Nil significant difference9 MonthssHNA proved useful in identifying red flag symptoms, and helping practice nurses decide when to seek further medical care for the patients. There was a high level of acceptability for patients and HCPs. However, integration of care did notoccur as intended because of problems linking hospital and general practice IT systemsEnhancing survivorship care planning for patients with localised prostate cancer using a couple-focused web-based, mHealth program: the results of a pilot feasibility study [26]Song et al. (2020), Journal of Cancer Survivorship Two-group randomised controlled pilot study62 Dyads, 31 control (SCP), 31 intervention group (SCP + PERC)Web-based mHealth programPERC (Patient Education Resources for Couples) designed for prostate cancer patients and their partners. It can be accessed on any device and includes modules on teamwork, managing treatment side effects, and promoting healthy behaviours. The program offers social support, including online forums, meetings with a health educator, and a resource centre31 patients in control group (SCP)Primary outcome: QOL (27-item Functional Assessment of Chronic Illness Therapy General Scale, FACT-G). Trend towards improved QOL with no statistical significance (FACT-G physical: 0.34, *p* = 0.27; social: 0.97, *p* = 0.08; emotional: 0.55, *p* = 0.16; and total score: 0.65, *p* = 0.35)6 months It is feasible to integrate existing web-based interventions, such as PERC, into standardised SCPs. Doing so improves patient symptom management and satisfaction, and enables fewer medical visits.A novel mHealth App (RyPros) for prostate cancer management: An accessibility and acceptability study [18]Wang (2021), Translational Andrology and UrologyPilot study; “accessibility and acceptability study”32 participants were enrolled, of whom 28 completed the 4-week follow-upmHealth ApplicationRyPros. mHeath app covering 4 domains: dynamic visualisation, reminders, assessments, and messagingNilPrimary Outcome: Not statedInferred1. Participation: 87.5% (28/32) 2. Acceptability:(64%) liked the app, and most participants (71%) were satisfied4 weeksIn this small pilot study, RyPros showed mostly positive user satisfaction, with the most useful domain the messaging function.Acceptability and usability of a patient portal for men with prostate cancer in follow-up care [27]O’Connor et al. (2022), Frontiers in Digital HealthMixed methods evaluationProstate cancer patients: sixty percent (1556/2599) of those who were eligible chose to register to use the portal Patient online web-based portal TrueNTH. Key functions include finding patient information, messaging the clinical team, checking PSA results, completing a Health MOT (Holistic needs assessment), and Patient educationNilPrimary Outcome: Not StatedInferredParticipation: Sixty percent of eligible patients registered to use the portal. Of these, 37% logged in at least once over a 6-month period and 52% over 12 months3 years with interviews over the course of 3 months after introductionA large proportion of participants found the patient portal acceptable and easy to use, with younger age correlating to registration (*p* < 0.001).Designing Tailored Internet Support to Assist Cancer Patients in Illness Management [28]Ruland et al. (2007), AMIA symposium ProceedingsRandomised Control TrialPlanned 320 breast and prostate cancer patients from throughout NorwayOnline computer interactive application WebChoice. An Interactive eHealth application developed to assist cancer patients in managing their illness. It comprises of 5 parts: (1) Assessment (2) Self-management (3) Patient information (4) Communication (5) Diary

1 yearN/AEvaluation of different features of an eHealth application for personalised illness management support: Cancer patients‚ use and appraisal of usefulness [17]Ruland (2013), IJMI320 breast and prostate cancer patients from throughout Norway, 162 in experimental group, of these 103 logged on at least twice and included as active users163 patients in control group (information sheet with suggestions for publicly available, cancer-relevant Internet sites)Primary outcome: Symptom distress thorugh Memorial Symptom Assessment ScaleYShort Form (MSAS-SF). Only significance was drop in Global Distress Index (*p* = 0.04)Patient usage and value of WebChoice varied. Most useful function was direct communication with nursing staff.
healthcare-12-01768-t002_Table 2Table 2Functionality of different eHealth technologies for prostate cancer shared care.ApplicationPROMMessaging FunctionEducation/Self-Care ToolsShared Care Plan Integration with Hospital EMRIntegration with Primary Health EMRImplementation StrategyCHAT-P
Produces RED-flag messages 



“Theoretically driven implementation strategy is required”sHNA (CHAT-P in GP)



Integration of care did not occur as intended because of problems linking hospital and general practice IT systems. 
e-OncoNote
GP and specialist messaging. No patient messaging function.



Difficulties in recruiting PROMIS CAT T
Instant messaging 



Hospital implementation strategy onlyeSYM
Instant messaging 



Multi-institutional hospital implementation strategy only Interaktor






PERC






RyPros
Instant messaging 




TrueNTH
E-mail style communication




WebChoice
Anonymous forum for group discussion




Notes: Adaptive tools and real-time alerts indicated as column titles. The extent to which these tools fulfilled the demands of shared care were colour-coded with a traffic light system (green: effective; orange: somewhat effective; red: not effective).

#### 3.4.3. Patient-Reported Outcome Measures (PROMs)

All platforms integrated patient-reported outcome measures (PROMs) into their eHealth interventions. Web-based patient portals, such as PROMIS CAT T (EPIC, MyChart), were used to collect patient-reported data on symptoms, quality of life, and care needs [19]. PROMs facilitated the identification of prevalent symptoms and allowed for a more patient-centred approach to care. The integration of PROMs into existing electronic health record systems enabled efficient data collection and facilitated clinical review when specific symptoms or care needs were identified.

#### 3.4.4. Communication Platforms

Effective communication between primary care providers, specialists, and patients is crucial in shared care. Differing communication strategies include the following: direct instant messaging, email-type message, and asynchronous communication. Interaktor has a messaging function through which patients can message nursing staff within business hours [14,15,22,23,24,25]. In contrast, eOncoNote, employs web- and text-based asynchronous systems to facilitate communication among healthcare professionals [13]. In general, integrated secure messaging and information sharing was highly valued by patient and clinicians [12,13,14,15,22,23,24,25]. By streamlining communication channels, these interventions aimed to enhance care coordination, cross-boundary continuity, and patient.-provider communication.

#### 3.4.5. Patient Education and Support

Many programs (PERC, CHAT P) include patient education and support for prostate cancer patients and their partners [26]. These programs offer modules on teamwork, managing treatment side effects, and promoting healthy behaviours. Additionally, they provide social support through online forums, meetings with health educators, and resource centres. By empowering patients and their partners with knowledge and support, these interventions aim to enhance patient self-management and satisfaction.

The studies showcase diverse ways to improve digital prostate cancer care, including holistic assessments, symptom tracking, and communication tools. These aim to enhance patient engagement, symptom control, and care quality. eHealth interventions with different designs and functions highlighted the role for multifaceted approaches for improving shared care.

### 3.5. Control Groups and Outcome Measures

Comprehensive meta-analysis on these studies was not plausible given variations in control groups and outcome measures (often limited or none). Comparator groups ranged widely from none to standard care to historical controls, causing further inconsistency. Most studies lacked clinical endpoints, relying instead on patient-reported outcomes or qualitative surveys, limiting assessment of health impacts (Table 1). Diverse study designs and small sample sizes further complicate comparisons. The heterogenicity of the various eHealth interventions and study durations introduced further difficulty making meaning extraction of data

## 4. Discussion

While many of these studies have part of a comprehensive shared care platform, their contributions to shared care are limited by their inability to address all realms. Effective implementation of eHealth in digital prostate cancer shared care is complex involving multisite, multidisciplinary integration. To successfully achieve this ambition, eHealth platforms need to facilitate all aspects of shared care. The central finding from this review is that current platforms fail to achieve comprehensive shared care facilitation. We dissect comparative enablers and barriers, strategies of platform development, implementation strategies, and platform strengths and weaknesses. We then formulate this to highlight gaps in contemporary practice.

### 4.1. Contrasting Enabling Strategies and Barriers

Previous studies reviewed present diverse eHealth interventions in aspects of digital prostate cancer shared care. They include the use of web-based platforms for holistic needs assessment [21], web-based asynchronous tools for communication between primary care providers and cancer specialists [13], integration of existing electronic health records for symptom screening and support [19], and the development of smartphone applications for symptom management [14].

### 4.2. Retrofitting Existing Systems vs. Creation of Novel Applications 

Two studies chose to leverage existing EMRs to facilitate their digital shared care platform [19,20]. Both studies used EPIC ^TM^ infrastructure to create new patient web-based patient portals (PROMIS CAT T) and patient applications (eSym). Whilst this overcomes intra-hospital integration barriers, it likely leads to less flexibility with primary care integration using various EMRs. In contrast to this, many other groups choose to custom-design novel web-portals and applications. 

### 4.3. Implementation Barriers 

The implementation of eHealth in digital prostate cancer shared care faces common barriers to any new eHealth platform. These include difficulties in integrating hospital and general practice IT systems [16], challenges in linking patients and healthcare teams through web-based portals [27], and varying levels of patient engagement with the applications [15]. 

However, implementation barriers are multifaceted, involving issues such as profitability, remuneration, accessibility, culture, and usability. Such socio-technical challenges and varying levels of technological readiness impact the implementation success of eHealth interventions [29,30]. 

Nanton et al. highlighted the difficulty of integrating web-based tools in prostate cancer care despite their potential benefits [12]. Petrovic’s trial found that eOncoNote reduced patient anxiety but did not improve care continuity, pointing to the need to evaluate both clinical and psychological outcomes [13].

Garcia et al. showed that integrating patient-reported outcomes into EHRs improved clinical reviews but increased patient anxiety, emphasizing the importance of psychological considerations [19]. Sundberg’s study on the Interaktor app demonstrated reduced symptoms during prostate cancer radiotherapy but required strong patient adherence and system integration [14]. Lastly, Clarke’s work on a shared digital platform revealed IT integration issues between hospitals and general practices, highlighting the need to address technical and systemic barriers for successful eHealth implementation [16].

Such implementation issues highlight the need for the planned effective coordination and interoperability of proposed eHealth interventions into existing healthcare digital architecture and clinical workflows. It is likely that effective implementation requires perceptible time savings, ease of use, and remuneration models for initial user implementation.

### 4.4. Within Hospitals or within Primary Care

Raising awareness and training staff on system use and patient introduction are crucial for implementation [19]. Avoiding burdensome workflows is also vital for staff uptake within larger care organisations.

Effective primary care consultation to inform uptake of new technologies is essential [31]. Without an effective model for uptake addressing key primary care concerns, rollout is limited as learning new systems is often seen as burdensome, costly, and time inefficient [32]. Robust primary care uptake of new shared care technology requires streamlined workflow (particularly intelligent synthesis of relevant information), improved communication, perceived patient benefit, and proper remuneration [31,32].

### 4.5. Common Strengths eHealth Applications: Why Move Away from Paper-Based Systems?

The reviewed studies demonstrate several strengths of eHealth applications in digital prostate cancer shared care (see functionality comparison in Table 2). These include improved patient-reported outcomes [12,13,16,17,18,19,20,25,26,27], enhanced communication between patients and healthcare providers [13,16,17,18,19,20,25,27], and increased patient engagement in symptom management. They offer a mixture of personalized care, provide real-time symptom tracking, and offer self-care advice and resources, thereby empowering patients to actively participate in their own care.

### 4.6. Common Weaknesses of eHealth Applications

Existing eHealth applications share certain weaknesses. These include implementation barriers, design flaws, lack of customization, and integration into existing electronic records. Existing publications on the use of digital tools in prostate cancer shared care show limited generalizability of findings due to small sample sizes and their pilot-type nature [15,18]. Additionally, some studies report the need for further refinement and customization of eHealth interventions to address individual patient needs and preferences [15]. Their clinical validation is further limited by a lack of clinical outcome measurements. Improved study designs with defined control groups and/or achievement-addressing scales are warranted. Future studies should target and mitigate for these common weaknesses early in their design phase.

### 4.7. Gaps and Limitations in Contemporary Practice

While existing eHealth platforms and their surrounding research have shown promise, there are significant gaps and limitations in eHealth prostate cancer shared care. These can be divided into application and study design factors.

#### 4.7.1. Application Factors

The eHealth platforms examined lacked comprehensive inclusion of all domains of shared care. They face significant common challenges related to integration with existing hospital and primary care EMRs. Future attempts to create shared care platforms need to ensure that an effective implementation strategy is in place prior to application rollout. Additionally, some studies lack an emphasis on patient-centred care and shared decision-making in the development and implementation of eHealth interventions. By involving patients in the design process, we can create interventions that better address their individual needs.

#### 4.7.2. Study Design Factors

Firstly, methodology variation used in the reviewed studies, including differences in study design, sample size, and duration, limits meaningful comparisons and definitive conclusions. To address this, large-scale randomized controlled trials and longer-term studies with defined clinical endpoints are required to establish the effectiveness (including oncological outcomes) and sustainability of eHealth interventions. Another important concern is the lack of standardized approaches and guidelines for developing and implementing eHealth platforms in digital prostate cancer shared care. The absence of consistent outcome measures and assessment tools makes it challenging to evaluate different interventions properly.

Furthermore, there is limited research on the cost-effectiveness and economic impact of eHealth applications in digital prostate cancer shared care. Future studies should consider the economic implications, feasibility, and long-term sustainability of implementing eHealth interventions in routine clinical practice.

The reviewed studies demonstrate that eHealth interventions, such as web-based platforms and smartphone applications, have the potential to enhance prostate cancer shared care by improving patient engagement, symptom management, and communication with healthcare providers. These tools empower patients to actively participate in their care, which is particularly valuable in managing chronic conditions like prostate cancer. However, the variability in study designs, populations, and outcome measures complicates the interpretation of results. The reliance on patient-reported outcomes and the lack of standardised endpoints make it difficult to assess the clinical effectiveness of these interventions. Despite these challenges, the studies highlight the promise of eHealth solutions in supporting personalized care. Future research should focus on standardising outcome measures and assessing the long-term impact of these digital tools to better understand their role in improving prostate cancer care.

## 5. Conclusions

This systematic review provides a comprehensive assessment of the role of eHealth in digital prostate cancer shared care. We identified ten eHealth platforms, which provided successful symptom tracking, needs assessment, and communications. Platform-based interventions improved some aspects of communication, symptom management, and care delivery. However, the ongoing clinical need for a robust digital platform that caters to all domains of shared care was identified.

The implementation of eHealth in digital prostate cancer shared care holds great potential for enhancing patient outcomes and improving the delivery of care. However, it requires careful consideration of strategies, addressing barriers, and refining eHealth applications to meet all patient’s share care needs. Future research should focus on standardization, larger-scale studies, long-term evaluation, cost-effectiveness, and patient-centred approaches to further advance the field of eHealth in digital prostate cancer shared care. As implementation has been hindered by difficulties engaging primary care, effective remuneration strategies to enable uptake will be essential in future digital prostate cancer shared care. Moreover, existing attempts at digital shared care have lacked interoperability with existing EMRs, which further hampers their practicability. Future studies would benefit from a focus on effect interoperability and implementation.

## Figures and Tables

**Figure 1 healthcare-12-01768-f001:**
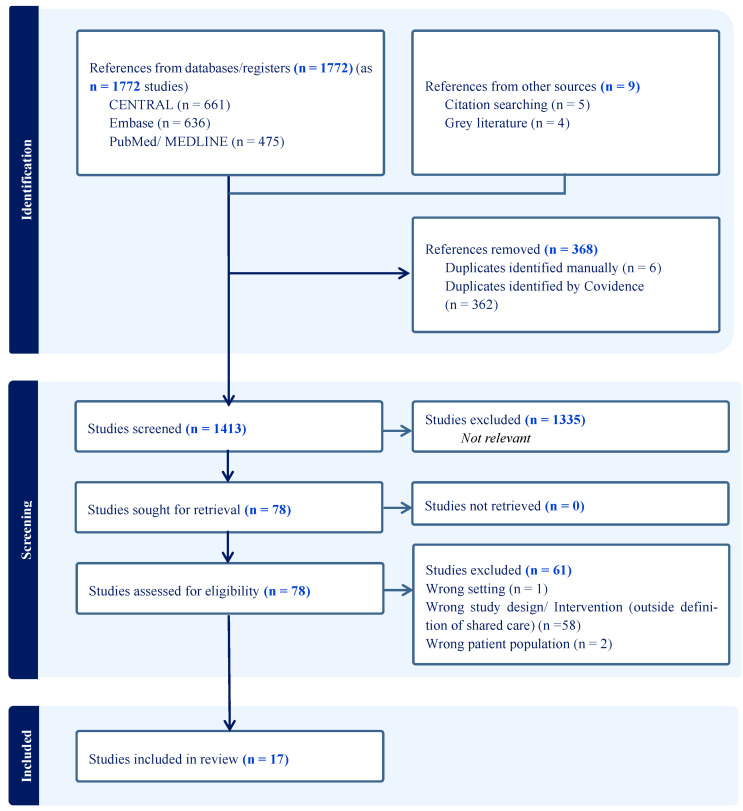
PRISMA search strategy (flowchart).

## Data Availability

All data generated and analysed in this study are presented in this manuscript.

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
