# Peer review of "eHealth Platforms Facilitate Prostate Cancer Shared Care: A Systematic Review"

_healthcare, 2024, doi:10.3390/healthcare12171768_

Round 1
Reviewer 1 Report
Comments and Suggestions for Authors
This paper explores the role of electronic health (eHealth) platforms in facilitating shared care for prostate cancer. The study screened relevant literature from databases such as PubMed, Embase, and the Cochrane Library, ultimately including 17 studies for review. The research found that various eHealth interventions (e.g., web platforms, applications, patient portals) could promote communication, symptom management, and comprehensive assessment, potentially helping to reduce anxiety, improve treatment outcomes, and increase patient engagement. The duration of the studies ranged from 9 months to 5 years, and the number of participants ranged from 16 to 3521. The research identified 10 types of eHealth platforms that achieved success in symptom tracking, needs assessment, and communication. However, it also highlighted a persistent clinical need for a robust digital platform that can address all aspects of shared care. The idea is novel, but there are still the following areas that need improvement:
1. The article should delve deeper into the barriers to implementing eHealth platforms and propose possible solutions or recommendations for improvement.
2. Discussion section: Expand the discussion section to include a broader interpretation of the results.
3. Conclusion section: Provide a more in-depth analysis, including a critical reflection on current practices and specific suggestions for future research directions.
Author Response
Thank you very much for this opportunity to revise our manuscript (Ref: Submission ID healthcare-3078777, " eHealth Platforms Facilitate Prostate Cancer Shared Care, A Systematic Review").
We thoroughly revised our manuscript according to the Reviewer’s and Editor’s comments.
The Rebuttal letter/document with our response to all comments is attached below to this re-submission letter.
Response to Reviewer 2
- Reviewer’s comment/question: The article should delve deeper into the barriers to implementing eHealth platforms and propose possible solutions or recommendations for improvement.
Author’s response: Thank you for this comment. We have amended the section on implementation in the discussion to address these comments (pages 18-19).
- Reviewer’s comment/question: Discussion section: Expand the discussion section to include a broader interpretation of the results.
Author’s response: We have added a paragraph on broader interpretation of results in the discussion to address this comment ( page 21).
- Reviewer’s comment/question: Conclusion section: Provide a more in-depth analysis, including a critical reflection on current practices and specific suggestions for future research directions.
Author’s response: Thank you for this comment. The conclusion has been amended ( page 21) to critically appraise current strategies and provide recommendations for future digital shared care prostate cancer studies.
We are looking forward to your favorable decision regarding our manuscript.
Sincerely,
Dr. Olga A. Sukocheva (on behalf of all co- authors)
Royal Adelaide Hospital, CALHN, Adelaide, Australia

Reviewer 2 Report
Comments and Suggestions for Authors
The authors present a valuable review-type study about: eHealth platforms facilitate shared care of prostate cancer
In order to raise the quality of the manuscript, the authors should explain in detail:
1. What motivated them to study prostate cancer survivors only.
2. Explain why they only chose the databases PubMed, Embase and Cochrane Libraries. They exclude Scopus, Clarivate, MDPI, etc.
3. The terms "eHealth" and "digital health" are relatively new. Why the authors choose to review studies published since 1/1/1946. It is necessary to know their arguments
4. The authors identified ten eHealth platforms that provided successful follow-up. It is important to disclose why the other platforms failed.
5. Expand the explanation of implementation barriers (studies on profitability, remuneration, accessibility, culture, usability and financial income)
6. Authors should consider removing the names of specific tasks (lines 137 and 138) and if they prefer to leave it, it is necessary to also present the tasks of the other authors. It is clearer in the Authors’ Contributions section
7. Finally, Table 2 is not explained. Table 2 appears after Table 1. But without explanation or definition of the colors. The only thing that is named is in section 4.5 (See functionality comparison in Table 2), which is very confusing.
Comments on the Quality of English LanguageSome text should be written in the past tense. Only results, discussion and conclusions in the present tense.
Author Response
Thank you very much for this opportunity to revise our manuscript (Ref: Submission ID healthcare-3078777, " eHealth Platforms Facilitate Prostate Cancer Shared Care, A Systematic Review").
We thoroughly revised our manuscript according to the Reviewer’s and Editor’s comments.
The Rebuttal letter/document with our response to all comments is attached below to this re-submission letter.
Response to Reviewer
- Reviewer’s comments/questions: What motivated them to study prostate cancer survivors only.
Author’s response: Thank you for your comment. We focused on prostate cancer survivors because they face unique and ongoing survivorship challenges, including managing treatment-related side effects and the risk of recurrence. Given that prostate cancer is one of the most common cancers in men, our goal was to explore digital solutions for survivorship care. The insights from this study aim to inform care strategies for future solutions which could be extended to other cancer survivor groups.
- Reviewer’s comments/questions: Explain why they only chose the databases PubMed, Embase and Cochrane Libraries. They exclude Scopus, Clarivate, MDPI, etc.
Author’s response: PubMed, Embase, and Cochrane libraries were chosen for this systematic review due to their strong focus on biomedical and healthcare literature, high-quality peer-reviewed content, and specialized indexing that enhances search precision. These databases are tailored to meet the needs of medical research, minimizing irrelevant results and avoiding redundancy. Broader databases like Scopus, Clarivate, and MDPI were excluded as they cover a wider range of disciplines, which can dilute the relevance and quality of search outcomes given the healthcare-focus of this study.
- The terms "eHealth" and "digital health" are relatively new. Why the authors choose to review studies published since 1/1/1946. It is necessary to know their arguments.
Author’s response: Thank you for this observation. We chose these dates to ensure all possible studies would be included in the review.
- The authors identified ten eHealth platforms that provided successful follow-up. It is important to disclose why the other platforms failed.
Author’s response: Thank you for the chance to clarify this. There were only ten platforms from the 17 studies that met the eligibility criteria of the search, inclusion and exclusion criteria. For instance, the main inclusion criteria were the description of the care model and original research status of the study.
- Expand the explanation of implementation barriers (studies on profitability, remuneration, accessibility, culture, usability and financial income).
Author’s response: Thank you for this comment. We have amended the section on implementation in the discussion to address these comments (pages 18-19,21).
- Authors should consider removing the names of specific tasks (lines 137 and 138) and if they prefer to leave it, it is necessary to also present the tasks of the other authors. It is clearer in the Authors' Contributions section
Author’s response: Thank you for this comment. This has been removed.
- Finally, Table 2 is not explained. Table 2 appears after Table 1. But without explanation or definition of the colors. The only thing that is named is in section 4.5 (See functionality comparison in Table 2), which is very confusing.
Author’s response: Thank you for this comment. This has been added with explanation in text (section 3.3., page 5).
We are looking forward to your favorable decision regarding our manuscript.
Sincerely,
Dr. Olga A. Sukocheva (on behalf of all co- authors)
Royal Adelaide Hospital, CALHN, Adelaide, Australia

Round 2
Reviewer 2 Report
Comments and Suggestions for Authors
This new revision highlights the efforts of the authors. They have revised and corrected the manuscript according to the reviewer's comments and observations. Accepted for publication in its current state.